# Learning a Metric without Supervision: Multimodal Registration using Synthetic Cycle Discrepancy

**Hanna Siebert**[*1]                                    SIEBERT@IMI.UNI-LUEBECK.DE

**Lasse Hansen**[*1]                                    HANSEN@IMI.UNI-LUEBECK.DE

**Mattias P. Heinrich**[*1]                               HEINRICH@IMI.UNI-LUEBECK.DE

[1] *Institute of Medical Informatics, Universität zu Lübeck, Germany*

## Abstract

Training deep learning based medical image registration methods involves the challenge of finding a suitable metric. To avoid the difficulty of choosing a metric for multimodal image registration, we propose a completely new concept relying on geometric instead of metric supervision with three-way registration cycles. Therefore, we create a synthetic image by applying a synthetic transformation on one of the input images. This leads to cycles that for each pair of input images comprise two multimodal transformations to be estimated and one known synthetic monomodal transformation. We minimise the discrepancy between the combined multimodal transformations and the synthetic monomodal transformation. By minimising this cycle discrepancy, we are able to learn multimodal registration between CT and MRI without metric supervision. Our method outperforms state-of-the-art metric supervision and comes very close to fully-supervised learning with ground truth labels.

**Keywords:** multimodal features, image registration, self-supervision.

## 1. Introduction

Especially for multimodal unsupervised deep learning based medical image registration methods no universally applicable similarity metric is available. In order to avoid the difficulty of choosing a metric for training, we propose a completely new concept. For learning multimodal features our method requires neither label supervision nor handcrafted metrics. The novel approach relies on geometric instead of metric supervision. Therefore, we introduce a cycle based concept. We aim to learn rigid multimodal registration between CT and MRI without metric supervision by minimising a cycle discrepancy. Our experimental validation on 3D rigid registration demonstrates the achievability of high accuracy and the simplicity of training such networks. Implementation details, open source code, and image data can be found at [github.com/multimodallearning/learning_without_metric](github.com/multimodallearning/learning_without_metric).

## 2. Methods

We introduce a learning concept for multimodal 3D image registration that learns without metric supervision. Therefore, we propose a method to learn with help of a self-supervised learning procedure using three-way cycles (Figure 1).

---

[*] Contributed equally

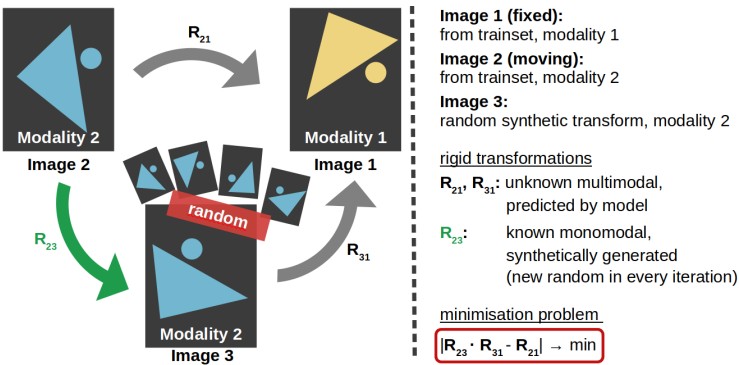

Figure 1: Our proposed learning concept aiming to minimise a cycle discrepancy.

Our method is based on geometric self-supervision by minimising the cycle discrepancy created through cycles consisting of two estimated multimodal and one known synthetic monomodal transformation. The transformation $R_{21}$ between a fixed image (1) and a moving image (2) is to be learned by our method. Within each training iteration, we deform the moving image by applying a known random transformation $R_{23}$ and hereby obtain a synthetic image (3). Combining the individual transformations leads to the minimisation problem of $|R_{23} \cdot R_{31} - R_{21}| \to \min$. For optimisation, we use the MSE loss function to minimise the discrepancy between the two flow fields generated by the matrices $R_{21}$ and $R_{23,31} = R_{23} \cdot R_{31}$. We create the synthetic transformations $R_{23}$ by randomly initialising rigid transformation matrices within an anatomically expected range.

The structure of our learning method comprises three main components for feature extraction, correlation, and registration. We chose to use a Y-shaped feature CNN architecture (Blendowski et al., 2021) with initially separate encoder blocks for each modality followed by shared weights within the last few layers. The obtained features are fed patch-wise into the correlation layer (w/o trainable weights) whose output is directly converted into displacement probabilities. We correlate the features by calculating patch-wise sum of squared differences and extract grid points with high similarity, which are used to define point-wise correspondences to calculate the rigid transformation matrix with a robust (trimmed) least squares fitting procedure. By using a dense correlation layer, large deformations can be captured robustly (Heinrich, 2019).

## 3. Experiments and Results

Our experiments are performed on 16 paired abdominal CT and MR scans from collections [1] [2] of the TCIA project (Clark et al., 2013). For evaluation, we have manually created labels for four abdominal organs (liver, spleen, left kidney, right kidney). The pre-processing comprises reorientation, resampling to an isotropic resolution of 2mm and cropping/padding to volume dimensions of $192 \times 160 \times 192$.

---

1. O Akin, P Elnajjar, M Heller, et al. Radiology data from the cancer genome atlas kidney renal clear cell carcinoma [tcga-kirc] collection. *The Cancer Imaging Archive*, 2016.
2. M Linehan, R Gautam, S Kirk, et al. Radiology data from the cancer genome atlas cervical kidney renal papillary cell carcinoma [kirp] collection. *The Cancer Imaging Archive*, 2016.

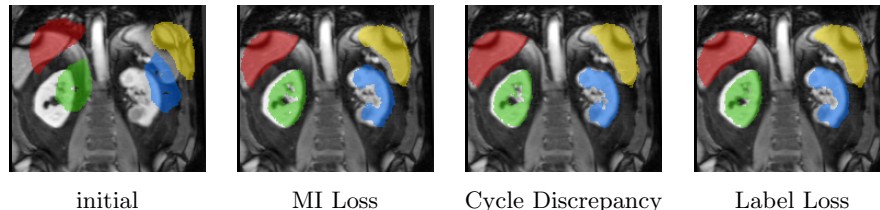

| initial | MI Loss | Cycle Discrepancy | Label Loss |

Figure 2: Qualitative results: fixed MRI and (warped) moving CT labels.

We compare our proposed self-supervised cycle learning with metric supervision (mutual information (MI)) and label supervision in a two-fold cross-validation. To increase the number of image pairs and model realistic variations in initial misalignment we augment the scans with 8 random rigid transformations each that on average reflect the same Dice overlap as the raw data. The results demonstrate a clear advantage of our proposed self-supervised learning without metric compared to the state-of-the-art MI metric loss (see Table 1 and Figure 2). This result nearly matches the results of full label supervision.

Table 1: Quantitative results: Dice scores listed by anatomical structures.

|  | liver 🟥 | spleen 🟨 | lkidney 🟦 | rkidney 🟩 | mean |
|---|---|---|---|---|---|
| initial | 59.3 | 36.9 | 36.6 | 37.0 | 43.0 |
| FeatCNN + MI Loss | 74.7 | 62.5 | 67.9 | 67.4 | 68.1 |
| FeatCNN + Cycle Discrepancy | 78.5 | 69.7 | 71.6 | 74.9 | 73.8 |
| FeatCNN + Label Loss | 79.3 | 71.1 | 76.0 | 75.0 | 75.3 |

## 4. Discussion and Outlook

We presented a new concept for multimodal feature learning without label supervision or handcrafted metrics. We trained modality independent feature extractors that enable highly accurate and fast rigid multimodal image alignment by minimising a cycle discrepancy. Various potential extensions might further improve our concepts including the the use of non-rigid deformation models, incremental/active learning for more meaningful synthetic transformations, fine-tuning at test time, and the combination with domain knowledge.

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
