# OpenReview forum: "Learning a Metric without Supervision: Multimodal Registration using Synthetic Cycle Discrepancy"
_MIDL.io/2021/Conference/Short — MIDL 2021 Poster_

### Official Review · Reviewer_ApvC · 2021-04-21

**Confidence:** 3
**Final Rating:** 3

**Summary:**

This paper presents an approach for multimodal registration, which is able to learn multimodal registration between
CT and MRI by minimizing a cycle discrepancy with self-supervised learning instead of a metric supervision. Evaluation is conducted on paired abdominal CT and MR scans with comparison to a metric supervision of mutual information and the label supervision.


**Strengths:**

The proposed method avoid the difficulty of choosing a metric for training multimodal registration based on deep learning. The results show that the method can approach the registration results obtained with full label supervision.

**Weaknesses:**

The method section only describes the general idea of the proposed registration framework, but how the whole framework and training process is realized is not clearly presented, such as what the correlation layer is and what the inputs and outputs of this layer are.

**Deanonymize Review:**

no

**Justification Of The Rating:**

The proposed method is reasonable and relaxes the need of choosing a metric supervision for training the deep network for multimodal registration. The registration results of abdominal CT and MRI images are also promising.

**Paper Type:**

methodological development

**Special Issue:**

no

---

### Official Review · Reviewer_3Hs2 · 2021-04-30

**Confidence:** 3
**Final Rating:** 3

**Summary:**

This paper proposes an unsupervised multimodal image registration method.  They first apply a known random transformation to generate a synthetic image from the fixed image, and then minimize the discrepancy between transformation from the fixed image and from the synthetic image. The description for the method can use some improvement. The experimental results show promising performance.

**Strengths:**

1. The proposed method is simple and light-weight, and the cycle discrepancy idea is interesting.
2. The graphical explanation is clear and helpful for understanding the method.
3. The experimental results show promising performance, close to full label supervision.

**Weaknesses:**

1. The discription in the third paragraph of 2. Method is not very clear. What is displacement probability? What are the features in the sentence below?

"We correlate the features by calculating patch-wise sum of squared differences and extract grid points with high similarity, which are used to define point-wise correspondences to calculate the rigid transformation matrix with a robust (trimmed) least squares fitting procedure."

2. In the minimization problem equation $| R_{23} \cdot R_{21} - R_{21} | \rightarrow min$, the moving image seems to be irrelevant and can be replaced by any other images? How is the method learning a transformance from the fixed image to the moving image?

3. Can you utilize more than one synthetic image in each iteration?

4. Can this method compare with general unsupervised correspondence learning methods?

**Deanonymize Review:**

no

**Justification Of The Rating:**

This paper proposes a simple but interesting unsupervised method for image registration with cycle discrepancy. While as described in weaknesses, the reviewer still have some questions concerning their method.

**Paper Type:**

methodological development

**Special Issue:**

no

---

### Meta-Review · Program_Chairs · 2021-05-09

**Recommendation:** Accept (Poster)
**Confidence:** 4

**Metareview:**

Both reviewers recommend to accept the paper but they point out some unclarities that should be addressed in the final version.

---

### Decision · Program_Chairs · 2021-05-11

Accept (Poster)